# At the roots of Plant Awareness Disparity (PAD): Semantic processing and numerosity perception

**Silvia Guerra** *, **Marco Roccato, Carolina Maria Oletto**, **Andrea Ghiani**, **Marco Bertamini**, **Luca Battaglini**

Department of General Psychology (DPG), University of Padova, Padova, Italy

* silvia.guerra@unipd.it

## Abstract

Plant Awareness Disparity (PAD) refers to the inability of humans to notice plants and recognize their importance. Among the various factors (e.g., cultural) contributing to PAD, the less prominent visual cues of plants (e.g., color) might be one of the main features making them less noticeable to human perception. Here, we investigated whether PAD affects basic numerosity perception, which represents a fundamental cognitive ability that allows individuals to interpret and interact with their surroundings. Across three experiments, we compared how participants perceive the numerosity of plants (specifically trees), animals, and minerals. Participants completed two tasks: an estimation task, in which they reported the exact number of items in a single set and a comparison task, which required them to discriminate numerosity between two sets of items. In Experiment 1, both tasks employed colored images. We hypothesized that participants would underestimate the number of plant items in comparison to animals and minerals, given that plant stimuli typically attract less attention. In Experiment 2, black and white images were used to test whether the green color of plants contributes to PAD. In Experiment 3, all items were rotated by 180° to disrupt semantic recognition and assess whether PAD arises from higher-level cognitive processes. Results revealed a consistent underestimation of plants in Experiment 1 and 2, but this effect diminished in Experiment 3. The reduction of this effect suggests that semantic recognition processes may contribute to PAD. These results highlight how cognitive biases toward plants can influence basic perceptual judgments essential for everyday functioning.

## Introduction

Plants represent more than 80% of the global biomass distribution of taxa on Earth [1]. Despite their ecological dominance, humans often overlook plants and underestimate their critical role in sustaining life. This lack of recognition has significant consequences for sustainability and conservation, impacting biodiversity. For example, nearly 40% of plant species are currently threatened with extinction, but because

**Data availability statement:** All data presented in the study are available at: https://osf.io/mzpr5/.

**Funding:** This work was supported by the Department of General Psychology (DPG) BATT_BIRD24_01 - Plant Awareness Project: Numerosity, Training and Brain Stimulation – PRID (Interdisciplinary Department Projects) granted to L.B and S.G.

**Competing interests:** The authors have declared that no competing interests exist.

plants receive less public attention than animals, they also receive significantly less conservation funding [2–7].

This human inability to perceive, understand and properly appreciate plants is known as Plant Awareness Disparity (PAD) [8–10]. Although PAD is still understudied, this phenomenon seems to capture three main dimensions: attention (i.e., the capacity to perceive, recognize, and distinguish plants from their surroundings), understanding (e.g., knowledge of plant biology and their contributions to ecosystems) and attitudes (i.e., emotional responses toward plants) [8–11]. Multiple aspects contribute to PAD such as educational, cultural [3,12–16], demographics [17–21] and cognitive factors [22]. A lack of knowledge about plants appears to be exacerbated by rapid technological change leading to a progressive loss of human–nature interactions [23–28]. Furthermore, education systems often exhibit a zoocentric bias by emphasising animals over plants. For example, science textbooks typically contain more content and images of animals than plants [29,30]. Similarly, scientists publish fewer papers about plants than animals in conservation journals [2,7]. This implicitly conveys the common misperceptions that plants are 'less important' than animals [31–35]. This view is reinforced by the fact that plants exhibit less 'active' behaviours compared to animals and less prominent visual cues than other organisms in the ecosystem [21,36–39]. Indeed, the morphological traits of plants, such as their green color and tendency to grow in dense aggregates contribute to a sense of environmental uniformity, which influences human visual perception [22]. Studies have shown that humans detect, remember and interpret images of animals more efficiently than images of plants in both simple [21,36] and complex scenes [40] and even when plants are observed moving on a human time scale [41]. These findings suggest that PAD is, in part, the result of differential processing of plants and animals. Nevertheless, the cognitive underpinnings of PAD remain largely unexplored.

To address this gap, we investigated the phenomenon of PAD during a numerosity discrimination task (i.e., perceiving the number of discrete items in a set [42]). Numerosity discrimination is a fundamental ability that helps an organism to understand its environment. It provides, indeed, important survival advantages by guiding the behaviour and decisions of different species of animals, including human beings [43–52] and plants [53–55]. For instance, the capacity to estimate the quantity of items (e.g., predators) in the environment, without serial counting, allows the organism to choose zones with more food, and quickly determine which group of competitors is more numerous [56]. With this in mind, we conducted three experiments in which participants were requested to observe different sets of images of plants (i.e., trees), animals (e.g., dog) or inanimate objects (i.e., minerals). Each image was presented without a background. Previous research on PAD has typically used images of plants embedded in environmental scenes [21,36,40], presented alongside other organisms (e.g., animals) [38,57], or video displaying clasping movement toward a pole performed by a pea plant [41]. However, to study visual responses to plants we chose to use them as segmented items, a format common to many visual search and numerosity perception tasks. In Experiment 1, participants were engaged in two tasks: i) estimation task: participants were requested to report the number of items in

an image and a ii) comparison task: a two-alternative forced-choice involving numerosity discrimination, in which participants were requested to select the stimulus containing a greater number of elements in the comparisons of stimuli within and between groups (i.e., animals, trees, and minerals). If plant stimuli tend to be given less attention and processed less efficiently [21,22,36,38,40,57], we expect judgements of the number of items belonging to the plant category to be less accurate and more variable than judgements of items belonging to the animal or mineral categories. That is, we predict that the number of items presented will be underestimated if they belong to the plant category. In Experiment 2, we used the same stimuli used in Experiment 1 but without color. We thought that if plant awareness disparity (PAD) stems primarily from the green color of the plants [22], by removing the color we should observe significantly enhanced attention to plants, and a decreasing of the PAD. Alternatively, if the number of items belonging to the plant category continues to be underestimated, this would suggest that PAD has a more semantic origin rather than being driven by the plant's color or other visual features. In Experiment 3, animals, plants (trees) and minerals were rotated through 180°. Object recognition is typically faster when objects are shown in canonical views, which align with stored mental representations [58,59]. In contrast, non-canonical orientations (i.e., rotating images) have been shown to disrupt recognition, as they require cognitive rotation to align with internal templates [60]. In line with this, we expect that the obstruction of semantic access to the three distinct categories would attenuate the results, given that unrecognised objects would exhibit a lack of perceptual salience.

In sum, we tested three key hypotheses across our studies: i) numerosity judgments for plant stimuli would be systematically underestimated compared to animals and minerals, reflecting PAD's influence on basic cognitive processes; ii) removing color information would reveal whether PAD stems primarily from visual features (specifically the green color of plants) or has deeper cognitive origins; iii) disrupting semantic recognition through image rotation would equalize performance across categories if PAD operates at a semantic level rather than a purely perceptual one. Altogether, our study would provide a novel investigation into whether and how plant blindness affects numerosity perception, a cognitive ability essential for environmental interaction.

## Materials and methods

### Participants

A total of 154 participants were recruited across the three experiments, each consisting of a comparison task and an estimation task. Each participant performed only one experiment. In Experiment 1 (Color), we recruited 51 participants (Mean age = 34.24 ± 17.34 years, age range = 19–76 years, 32 females), of which 2 participants only completed the estimation task, while 3 participants only completed the comparison task. In Experiment 2 (Black and White), 51 participants were recruited (Mean age = 33.14 ± 14.07 years, age range = 20–70 years, 33 females), of which 3 participants only completed the estimation task, while 2 participants only completed the comparison task. In Experiment 3 (Upside down), we recruited 52 participants (Mean age = 28.33 ± 10.75 years, age range = 20–58 years, 24 females), of which 7 participants only completed the estimation task, while 3 participants only completed the comparison task. The recruitment period for this study was performed from the 10th of February 2025–1st August 2025. All participants provided their written informed consent before the beginning of each online experimental session. The experimental protocol was approved by the local institutional review board at the Department of General Psychology of the University of Padova (protocol number, 981-a; approved on 22nd December 2024) and was conducted in accordance with the ethical standards laid out in the Declaration of Helsinki.

### Experimental stimuli

The experimental stimuli were images of animals (e.g., cat), minerals and plants (i.e., trees; Fig 1A shows all the possible stimulus types). The images were selected from the internet to gather three exemplars belonging to three semantic

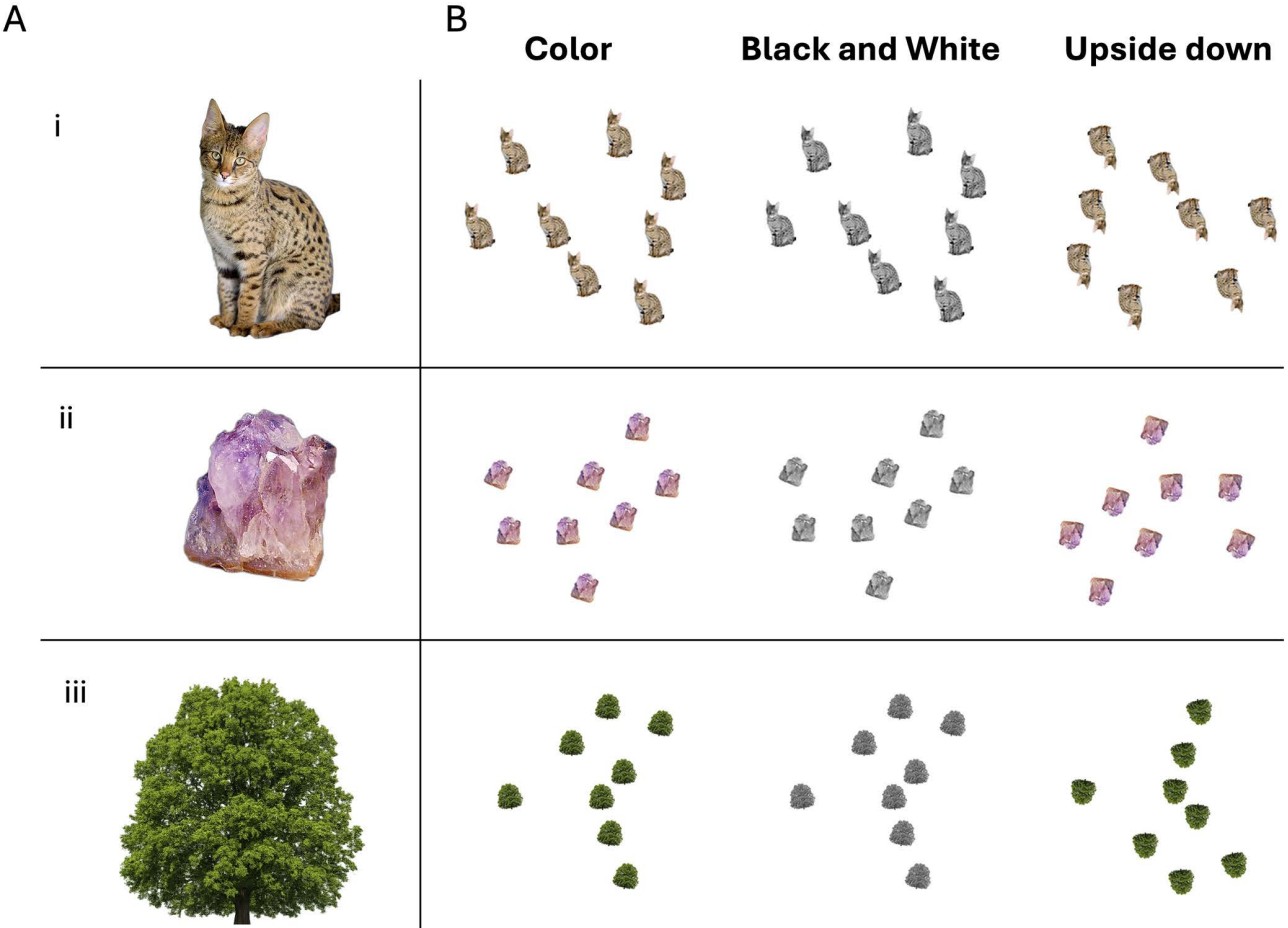

**Fig 1. Examples of experimental stimuli belonging to the animal (i), mineral (ii), and plant (iii) categories (A).** The three stimulus variants (Color, Black and White, and Upside-down) used in the three experiments are shown, using an example stimulus composed of 8 items **(B)**. Image sources: animal: Savannah cat portrait photograph by Jason Douglas (2006), Public Domain, Wikimedia Commons; mineral: Amethyst 3353 photograph by Dave Dyet (2007), Public Domain, Wikimedia Commons; Representative plant stimulus similar to those used in the study: Oak tree with memorial bench, Croxley Green, Herts (May) photograph by Hogweard (2011), Public Domain, Wikimedia Commons. Original plant stimuli are not displayed due to copyright restrictions.

categories: animals, minerals and trees. Animal and mineral images were collected from Wikimedia Commons (https://commons.wikimedia.org/), while tree images were collected from Google Images. Each source image depicted a single category exemplar on a varied background, which was subsequently removed to isolate the foreground object (Fig 1A).

Each stimulus used in the experiment was created by arranging multiple copies of a specific category exemplar (e.g., a stimulus could contain 7 copies of the cat, while another could contain 8 copies of the quartz rock). To study perception of numerosity outside the typical subitizing limit (i.e., the rapid and accurate perceptual enumeration of small quantities, typically 1–4 items, without the need for counting, and occurs even under very brief stimulus presentations), our displays had 7, 8 or 9 items. Items were always exemplars from the same category (Fig 1B). For each stimulus, the corresponding number of copies was positioned within an invisible circular area with a radius of 200 pixels, centered on a white background.

To control for low-level visual features, the surface area (in pixels) of each individual copy was held constant across all categories, exemplars, and numerosity levels. Furthermore, to ensure copies could be positioned to avoid overlap without

creating a grid-like configuration, each underwent a random rotational transformation. The rotation angle for each copy was independently drawn from a uniform distribution ranging from −15 to +15 degrees. This arrangement not only diversified the visual presentation but was also intended to enhance the overall difficulty of the tasks by introducing variability in the orientation of each element. Three versions of the stimulus set were prepared. The first version (i.e., Color) consisted of colored images. The second version (i.e., Black and White) removed color from the stimuli. The third version (i.e., Upside down) retained color but images were rotated 180° compared to the original version (Fig 1B). Coordinates of the item copies used for each arrangement were kept fixed across the stimulus sets, so that only colour or orientation would be manipulated.

The choice of item size was based on a preliminary pilot study. Although the Weber fraction for discriminating numerosity in the chosen conditions (set sizes 7, 8, 9) indicates that the task is relatively difficult [61], this difficulty is intentional and functional to our research objectives. Indeed, if the task were too easy, Plant Awareness Disparity (PAD) might not manifest in the results. Our hypothesis is that PAD emerges more clearly under conditions of relative perceptual uncertainty, when the cognitive system must rely more heavily on higher-level processing. In these conditions of moderate difficulty, attentional and perceptual biases toward different semantic categories are more likely to influence numerical judgment. Therefore, the choice to use closely spaced numerosity (7, 8, 9) was deliberate to create conditions where discrimination requires sufficient cognitive engagement to reveal the effect of stimulus category.

## Procedure

Two tasks were administered online using the jsPsych library (v7.3.4), a JavaScript-based framework for creating and running behavioural experiments in a web browser [62], and a departmental JATOS server (v3.6.1) [63] to manage data collection. Participants accessed the tasks on their personal desktop or laptop computers and received instructions on how to correctly perform the tasks remotely. Before starting each task, participants were presented with a language selection screen (i.e., Italian or English), followed by a digital informed consent form screen and a brief demographic questionnaire collecting their self-generated participant code, age, and sex information.

**Numerosity estimation.** The first task was a numerosity estimation task. On each trial, participants were asked to accurately report the number of items presented in a briefly displayed visual array. The task began with a practice block consisting of 20 trials, followed by the main experimental session. Stimuli for the practice trials were randomly sampled from the full set of stimuli that would occur in the subsequent experimental blocks. During the practice block only, after each response, participants received feedback for 1500 ms indicating whether their answer was correct or incorrect. Subsequently, participants completed 4 experimental blocks of 54 trials each, for a total of 216 experimental trials. At the end of each experimental block – except the last – a message invited participants to take a short break and resume the task when they were ready. Each trial began with a 1000 ms fixation cross, followed by a 100 ms stimulus image showing several items. A 400 ms circular white-noise mask then appeared to prevent afterimages, followed by a 400 ms blank screen (Fig 2A). The circular white-noise mask had the same radius as the circular area in which the items were presented (i.e., 200 pixels). Finally, a response screen prompted participants to type the number of items they saw using the keyboard. Responses had no time limit and were submitted by pressing the Enter key (Fig 2A). No feedback on performance was provided during the experimental blocks.

The stimuli for this task were generated by combining three factors with three levels each: semantic category (i.e., animal, mineral, tree), specific category exemplar (e.g., cat, dog, or goldfish within the 'animal' category), and 3 numerosity levels (i.e., 7, 8, or 9 items). This resulted in 27 unique stimulus conditions. For the main experiment, each of these conditions was repeated 8 times, leading to a total of 216 trials (3 categories × 3 exemplars × 3 numerosity × 8 repetitions). The order of trial presentation was fully randomized for each participant. The dependent variable of interest was the numerical value entered by the participant on each trial to indicate perceived numerosity.

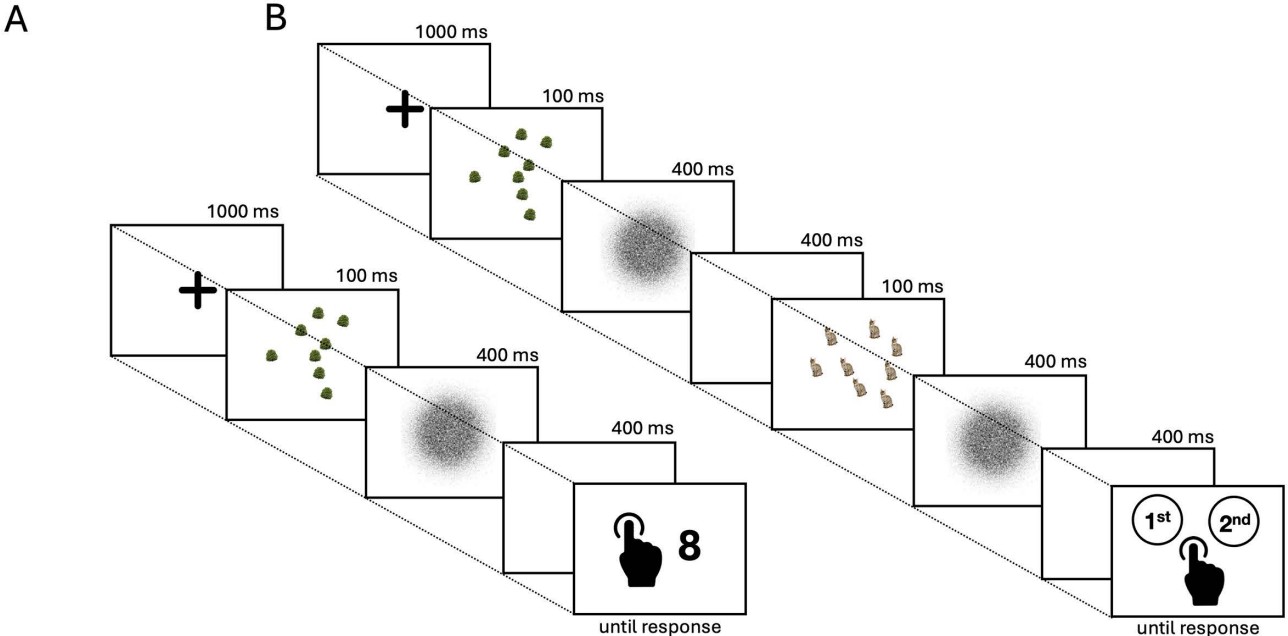

**Fig 2.** *Estimation task*: a fixation cross was shown for 1000 ms until the appearance of the stimulus for 100 ms. A 400 ms white noise mask was shown immediately after the stimulus to prevent aftereffects, followed by a 400 ms blank screen. Participants could then input their response through the keyboard **(A)**. *Comparison task*: after viewing a fixation cross for 1000 ms, participants were shown two intervals containing a 100 ms stimulus each. The two stimuli always differed by category and numerosity. Immediately after each stimulus, participants saw a 400 ms white noise mask followed by a 400 ms blank screen. After the end of the second blank screen, participants could then input their response on a keyboard, indicating which interval contained the highest number of items **(B)**. Image sources: animal: Savannah cat portrait photograph by Jason Douglas (2006), Public Domain, Wikimedia Commons; Representative plant stimulus similar to those used in the study: Oak tree with memorial bench, Croxley Green, Herts (May) photograph by Hogweard (2011), Public Domain, Wikimedia Commons. Original plant stimuli are not displayed due to copyright restrictions.

**Numerosity comparison.** The second task was a two-alternative forced-choice (2AFC) numerosity comparison task. During each trial, participants were required to judge which of two sequential images, briefly displayed, contained the greater number of items. The overall structure of this task mirrored the estimation task. The task began with a 20-trial practice block, followed by 4 experimental blocks of 54 trials each (216 trials in total). Instructions were displayed before the practice block and each experimental block, and self-paced breaks were offered between blocks. The 20 practice trials were a random subset sampled without replacement from these 216 trials. During the practice trials, feedback on whether the answer was correct or incorrect was provided for 1500 ms after each response. No feedback was given during the main experimental blocks. Each trial consisted of a central fixation cross displayed for 1000 ms. Then, the first stimulus image (S1) was presented for 100 ms, followed by a 400 ms white-noise mask. After the first mask, a blank screen was shown for a 400 ms inter-stimulus interval (ISI). After the ISI, the second stimulus image (S2) was presented for 100 ms, followed by a second 400 ms white-noise mask. After the second mask, a final blank screen was shown for 400 ms. Finally, a response screen appeared, asking to report which of the two stimuli was perceived to contain the highest number of items (Fig 2B). Participants were instructed to press the 'A' key if they believed the first image (S1) presented more items and the 'L' key if they believed the second image (S2) presented more items (Fig 2B). Responses were self-paced. The comparison task investigates performance on cross-categorical judgments involving trees. The design systematically paired stimuli from the 'plant' category (i.e., trees) with stimuli from the 'animal' and 'mineral' categories. All trials showed a stimulus belonging to the 'plant' category and a stimulus belonging to the 'non-plant' category (i.e., 'animal' or 'mineral'). The presentation order was counterbalanced, resulting in four category pairing conditions: tree-animal,

animal-tree, tree-mineral, and mineral-tree. Within each of these pairings, all 9 possible combinations of the 3 exemplars from each category were used (e.g., for tree-animal, tree exemplar 1 was paired with animal exemplars 1, 2, and 3; tree exemplar 2 with animal exemplars 1, 2, and 3, etc.). Furthermore, the numerosity of the two stimuli were drawn from three specific pairs (i.e., 7 vs. 8, 7 vs. 9, and 8 vs. 9). To account for order, assignment of the highest numerosity to each interval was also counterbalanced, resulting in 6 numerosity pairings: (7, 8), (8, 7), (7, 9), (9, 7), (8, 9), and (9, 8), where the first number corresponds to S1 and the second to S2. The combination of these factors resulted in 216 unique trials (4 category pairings×9 exemplar combinations×6 numerosity pairings). The trial order was fully randomized for each participant. The dependent variable of interest was the participant's choice of which stimulus had the highest numerosity.

## Data analysis

Data were analyzed in the R environment (Version 4.3.0) after removal of practice trials. For the estimation dataset, responses below 4 were excluded to eliminate trials in which no stimuli were perceived due to technical problems or inattentiveness (response = 0) and to avoid the subitizing range (1–3). Since the stimuli always contained 7, 8, or 9 items, responses falling within the subitizing range would imply a gross misperception of the display, making such errors extremely unlikely under normal task engagement. Responses above 12 were also excluded, thus adopting a symmetric exclusion range around the true numerosity. This criterion also removed responses that were most likely typos (e.g., typing 77 instead of 7), which would otherwise yield abnormally large estimation error values. Such typos were in fact permitted by the two-digit input box, to avoid revealing that the maximum true numerosity level was set at 9. In the estimation dataset, 750 outlier trials were removed in this way (2.38% of total trials). Supplementary analyses using different outlier rejection methods, in which residual normality and overdispersion were also checked, are available at https://osf.io/mzpr5/. These analyses show that core results are consistent across methods. We then calculated the estimation error using the formula: error = number typed – trial numerosity. A negative error corresponds to an underestimation of the number of elements per trial, while a positive error corresponds to an overestimation. We then performed a mixed effect model using the error as the dependent variable, and category, experiment version (Color, Black and White, or Upside down), and numerosity as the independent fixed factors. For the comparison dataset, we split the data into 'tree vs animal' and 'tree vs mineral' comparisons. We then performed two binomial mixed effect models using accuracy as the dependent variable, and the category with the highest numerosity and the experiment version as independent fixed factors. The random factor was the participant. Global tests of fixed effects (main effects and interactions) were obtained as Type III Wald $\chi^2$ statistics using the Anova() function from the *car* package. Throughout the following section, mentions of 'ANOVA' refer to these tests on the mixed models. Significant omnibus effects were followed up with estimated marginal means and Tukey-adjusted pairwise contrasts computed with the *emmeans* package.

## Results

Participants completed two tasks across three experiment versions: (i) numerosity estimation task, in which they reported the exact number of items in a single set) and (ii) numerosity comparison task, in which they discriminated between the numerosity of two sets of items. Experiment 1. We tested the hypothesis that plant stimuli would be processed less efficiently than animal and mineral stimuli, as these stimuli typically receive less attention [21,22,36,38,40,57]. Based on this, we expected lower accuracy and greater variability in numerosity judgments for the plant category. Experiment 2. We examined whether plant awareness disparity (PAD) is primarily driven by visual features, specifically the green color of plants. Using the same stimuli in black-and-white, we predicted that if PAD depends on color, removing it would enhance attention and reduce the effect. Conversely, a persistent underestimation of plant numerosity would suggest a more semantic origin of PAD. Experiment 3. We tested whether limiting semantic access would reduce category differences. Stimuli were rotated by 180°, a manipulation known to disrupt recognition. We expected this manipulation to attenuate differences between categories, as reduced recognizability should diminish perceptual salience across all stimulus types.

## Numerosity estimation results

The ANOVA for the estimation task revealed significant main effects of category ($\chi^2 = 396.61$, df = 2, p < .001), experiment version ($\chi^2 = 13.05$, df = 2, p = .001), and numerosity ($\chi^2 = 4169.14$, df = 2, p < .001). All two-way interactions were significant: category × experiment version ($\chi^2 = 85.20$, df = 4, p < .001), category × numerosity ($\chi^2 = 31.82$, df = 4, p < .001), and experiment version × numerosity ($\chi^2 = 74.51$, df = 4, p < .001), as well as the three-way interaction category × experiment version × numerosity ($\chi^2 = 16.16$, df = 8, p = .040). Fig 3A shows the average error in the estimation task.

**Pairwise comparisons across categories.** Pairwise comparisons across categories (averaged over numerosity and experiment version) showed a significant difference between animal and tree (estimate = 0.21, t = 19.80, p < .001), mineral and tree (estimate = 0.09, t = 8.52, p < .001), and animal and mineral (estimate = 0.12, t = 11.30, p < .001).

**Pairwise comparisons across experiment version.** Pairwise comparisons across experiment version (averaged over category and numerosity) revealed a difference between the Upside down and Color (estimate = 0.29, t = 2.60, p = .028) and Upside down and Black and White experiments (estimate = 0.39, t = 3.48, p = .002), but there was no significant difference between Color and Black and White versions (estimate = −0.10, t = −0.87, p = .662).

**Pairwise comparisons across experiment version and categories.** Pairwise comparisons across experiments (averaged over numerosity) revealed a difference between tree and animal (estimate = 0.27, t = 14.41, p < .001), tree and mineral (estimate = 0.14, t = 7.77, p < .001) and animal and mineral (estimate = 0.12, t = 6.65, p < .001) in the Color experiment. In the Black and White experiment, there was a significant difference between animal and tree (estimate = 0.29, t = 15.84, p < .001), mineral and tree (estimate = 0.12, t = 6.76, p < .001), and animal and mineral (estimate = 0.17, t = 9.10, p < .001). In the Upside down experiment, there was a significant difference between animal and tree (estimate = 0.08, t = 4.08, p < .001), and between animal and mineral (estimate = 0.07, t = 3.83, p < .001), but no difference between mineral and tree (estimate = 0.01, t = 0.25, p = .965).

**Pairwise comparisons across numerosity.** Pairwise comparisons across numerosity (averaged over category and experiment version) showed significant differences between numerosity 7 and 8 (estimate = 0.25, t = 23.47, p < .001), numerosity 7 and 9 (estimate = 0.68, t = 63.88, p < .001), and numerosity 8 and 9 (estimate = 0.43, t = 40.42, p < .001).

## Numerosity comparison results

**Tree vs animal.** The ANOVA for the tree vs animal comparison revealed a main effect of category ($\chi^2 = 579.41$, df = 1, p < .001) and of the interaction category × experiment version ($\chi^2 = 82.90$, df = 2, p < .001), but no significant effect of experiment version ($\chi^2 = 5.27$, df = 2, p = .072). Fig 3B shows the mean accuracy in the comparison task when the selected category was the one with higher numerosity.

**Pairwise comparisons across categories.** Pairwise comparisons across categories revealed a significant difference between animal and tree categories in all experiment versions (Color: odds ratio = 4.10, z-ratio = 18.63, p < .001; Black and White: odds ratio = 3.19, z-ratio = 16.49, p < .001; Upside down: odds ratio = 1.63, z-ratio = 6.666, p < .001).

**Pairwise comparisons across categories and experiment version.** Pairwise comparisons revealed a significant difference in the tree category between the Color and Upside down (odds ratio = 0.64, z-ratio = −3.83, p < .001) and between the Black and White and Upside down experiments (odds ratio = 0.59, z-ratio = −4.51, p < .001) but not between the Color and the Black and White version (odds ratio = 1.08, z-ratio = 0.72, p = .751). For the animal category, pairwise comparisons revealed a significant difference between Color and the Black and White experiment (odds ratio = 1.39, z-ratio = 2.586, p = .026), and between Color and the Upside down experiments (odds ratio = 1.61, z-ratio = 3.689, p = .001), but no significant difference between Black and White and Upside down version (odds ratio = 1.16, z-ratio = 1.144, p = .487).

**Mineral vs tree.** The ANOVA for the mineral vs tree comparison revealed no effect of category ($\chi^2 = 3.08$, df = 1, p = .079), and experiment version ($\chi^2 = 4.12$, df = 2, p = .128), but a significant interaction category × experiment version ($\chi^2 = 50.91$, df = 2, p < .001).

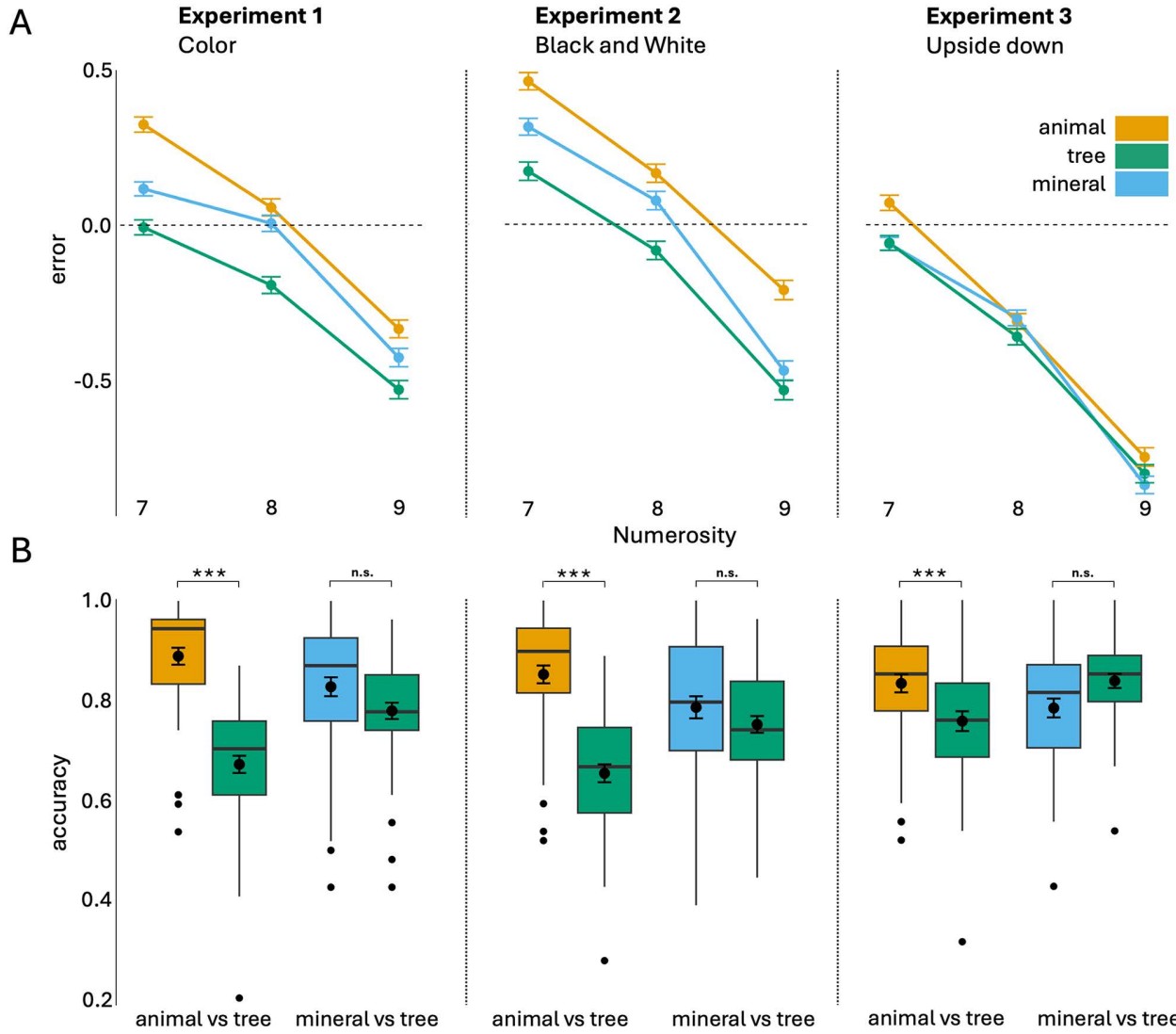

**Fig 3. Error in the *estimation task* shown separately for each numerosity.** Error bars represent the Standard Error of the Mean (SEM) **(A)**. Estimation data were analyzed using a mixed-effects model with error as the dependent variable, category, experiment version (Color, Black and White, Upside-down), and numerosity as fixed effects, and participant as a random effect. Mean accuracy in the *comparison task* when the selected category had the higher numerosity. Error bars represent SEM **(B)**. Comparison data were analyzed using binomial mixed-effects models separately for tree vs. animal and tree vs. mineral comparisons, with accuracy as the dependent variable, highest-numerosity category and experiment version as fixed effects, and participant as a random effect. Global effects were assessed using Type III Wald χ² tests. *** refers to $p < .001$; *n.s.* refers to not significant.

**Pairwise comparisons across experiment version.** Pairwise comparisons revealed a significant difference in the tree category between Color and Upside down (odds ratio = 0.667, z-ratio = −3.03, p = .007) and between the Black and White and Upside down experiments (odds ratio = 0.581, z-ratio = −4.05, p < .001), but no significant difference between the Color and Black and White version (odds ratio = 1.147, z-ratio = 1.069, p = .533). For the mineral category, pairwise comparisons revealed no significant difference between Color and Upside down experiment (odds ratio = 1.330, z-ratio = 2.140, p = .082), between Color and Black and White (odds ratio = 1.287, z-ratio = 1.923, p = .132), and between Black and White and Upside down experiments (odds ratio = 1.033, z-ratio = 0.246, p = .967). Pairwise comparisons across

experiment version revealed a significant difference between mineral and tree categories in all experiments (Color: odds ratio = 1.38, z-ratio = 4.525, p < .001; Black and White: odds ratio = 1.23, z-ratio = 3.025, p = .003; Upside down: odds ratio = 0.69, z-ratio = −4.936, p < .001).

## Discussion

The present study investigated whether Plant Awareness Disparity (PAD) affects basic numerosity perception, a fundamental cognitive ability that allows individuals to interpret and interact with their surroundings. Across three experiments, we examined how participants perceive and estimate the numerosity of plants (specifically trees), animals, and minerals under different visual conditions. Our results revealed a consistent underestimation of plants compared to animals and minerals in both Experiment 1 (Color) and Experiment 2 (Black and White). Specifically, in the estimation task, participants systematically reported lower numbers when counting plants compared to animals or minerals, despite all stimuli containing the same number of items. Additionally, in the comparison task, participants showed reduced accuracy when comparing trees versus animals or minerals, particularly when the tree category contained the higher numerosity. Most importantly, these effects were significantly reduced in Experiment 3 (Upside down), where stimuli were rotated 180° to disrupt semantic recognition. Although differences between animals and trees remained significant in this condition, they were considerably smaller. Most notably, the distinction between minerals and trees disappeared entirely. These results suggest that when semantic recognition is impaired, the perceptual bias towards plants is reduced. To our knowledge, this is the first study reporting that PAD influences even basic numerosity judgments, supporting the hypothesis that this phenomenon operates at a fundamental cognitive level.

Our findings are consistent with previous studies indicating that plants are less noticeable to the human eye [22]. Indeed, participants underestimated the number of plants compared to animals and minerals, regardless of how many items were presented. Notably, this underestimation persisted even when the items were shown in the Black and White version, suggesting that PAD is not primarily driven by the green color of plants. Therefore, plant features such as color uniformity, previously considered one of the main factors reducing their salience, do not appear to significantly affect how humans process visual information. Instead, the significant reduction of the plant underestimation effect with rotated stimuli suggests that PAD may arise from higher-level cognitive processes related to semantic recognition. When semantic access was impaired through rotation, the disparity between plants and other categories, particularly minerals, was substantially reduced. These findings align with theories suggesting that PAD stems from how humans categorize and process plant information at a conceptual level [8,10]. It has, indeed, been proposed that the human brain organizes conceptual knowledge into distinct categories such as animals, plants, and artifacts [22,64]. People detect animals' items (i.e., animates) more quickly and more frequently than those involving plants, or other inanimate objects such as vehicles, and tools [38]. Moreover, animate items tend to be remembered better than inanimate ones [65]. However, investigations comparing animates and inanimates have not directly contrasted plants and animals, often relying instead on fruits [66] or flowers [67] as representative plant stimuli. Our research provides novel evidence about how conceptual knowledge may be structured by directly comparing items from the animal, mineral, and plant categories. Consistent with previous findings, our results showed that animate items (i.e., animals) are judged more effectively than inanimate objects (i.e., minerals) [3,9,10,38,57,68–70]. Additionally, our findings revealed that minerals were estimated more accurately than plants, suggesting that plants may lack distinctive perceptual or conceptual features that enable rapid and accurate recognition. These results further support previous evidence highlighting the lower position of plants on the animacy spectrum, which may contribute to humans' tendency to undervalue them [3,7,9,10,38,57,68–70]. A plausible interpretation of these findings lies in the framework of the evolutionary hypothesis. The observed cognitive bias against plants in numerosity tasks may reflect evolutionary adaptations. Throughout human evolution, animals presented both immediate threats and food resources requiring rapid detection and quantification, while plants generally posed fewer immediate dangers [38]. Furthermore, the absence of visible movement in plants may cause them to be perceived as inanimate objects, reducing their

ability to capture human attention. Indeed, when plant movement is presented on a time scale comparable to our own, our awareness and appreciation of the green kingdom increase [41]. This evolutionary perspective might explain why animals consistently received more accurate numerosity judgments across all conditions. Another possible explanation could be rooted in cultural and educational factors, which may play a significant role in shaping the PAD phenomenon. Prior knowledge affects perceptual attention not only by making a stimulus more meaningful, but also by helping us interpret the stimulus and its components and encode it into memory [71]. The zoocentric bias in education systems and cultural representations likely reinforces cognitive patterns that diminish attention to and processing of plant stimuli [29,30]. When semantic recognition was disrupted through image rotation, these learned biases were partially circumvented, resulting in more equitable processing across categories.

The present findings have several important implications. First, they demonstrate that PAD extends beyond explicit attitudes or knowledge deficits to influence basic perceptual and cognitive processes. This suggests that efforts to address PAD must consider not only educational interventions but also how plants are represented and processed at a cognitive level. Second, our results highlight the role of semantic processing in PAD, suggesting that interventions targeting conceptual representations of plants might be particularly effective. Educational approaches that enhance the salience and significance of plant stimuli could potentially reduce perceptual biases against plants. Finally, the persistence of PAD's effects across the Color and Black and White experiments, but the reduction in the Upside down version, provides valuable insights into the cognitive mechanisms underlying this phenomenon. This knowledge can inform more targeted interventions to address plant awareness disparity in educational, conservation, and public awareness contexts.

Several limitations should be considered when interpreting our findings. First, our stimuli consisted of isolated images without ecological context, which may not fully capture how plants are perceived in natural environments. Future research could examine whether similar effects occur with more naturalistic stimuli or in real-world settings. Second, while our study focused on trees as representatives of the plant category, plants encompass a vast diversity of forms. Further research could investigate whether similar effects occur with different plant types, such as flowers, grasses, or shrubs, which might vary in their visual salience and cultural significance. Third, enhanced PAD may be associated with growing up in urban areas, which has been linked to a reduced connection to nature. Future research could incorporate an ad hoc PAD questionnaire to assess participants' sensitivity to the role and importance of plants for human life and other living organisms [72–75]. Including this measure would provide a more comprehensive understanding of the other factors influencing plant awareness. Fourth, our participant sample was limited to adults with access to online experiments. Future studies should explore whether similar effects are present in different age groups, particularly children, whose plant awareness might be more malleable and responsive to intervention. Future research could also explore potential interventions to reduce PAD at the perceptual level. For instance, training paradigms that enhance attention to plant features or emphasize the ecological significance of plants might mitigate perceptual biases. Additionally, neuroimaging studies could provide insights into the neural correlates of PAD, potentially identifying specific processing pathways that contribute to this phenomenon. Moreover, we acknowledge that at the moment it is difficult to decide between perceptual and cognitive accounts of PAD. While numerosity perception was employed to investigate the phenomenon at a basic level, the fact that the 'Upside down' condition yielded a weaker effect does not definitively establish the origin of the effect. Disambiguating between perceptual and non-perceptual (cognitive) mechanisms at this stage is challenging. However, it is important to emphasize that our study represents the first attempt to investigate PAD using a numerosity paradigm, providing evidence that this phenomenon influences basic numerical judgments. Our research team is currently developing experimental designs specifically aimed at addressing this perceptual-cognitive distinction. Definitive answers may emerge from more data and successful replications across different laboratories. This work thus serves as a foundation for a deeper understanding of the mechanisms underlying PAD.

Finally, we consider it important to point out that there is widespread consensus regarding the necessity of power analysis before conducting experiments. While we addressed this consideration prior to data collection, we had concerns

about the reliability of effect sizes as there are no similar studies in literature to rely on. We recognize that power analysis has limitations; its accuracy depends on reliable estimates of expected effect size, data variability, and other study parameters. Inaccurate estimates can compromise the power analysis validity. Considering these factors, and knowing that repeated measures studies with small effect sizes typically require fewer than 50 participants, we ultimately collected data from about 50 participants per experiment.

## Conclusion

The Plant Awareness Disparity (PAD) is a complex phenomenon shaped by well-recognized cultural and educational factors. In this study, we contribute new evidence on the cognitive underpinnings of PAD, showing that it influences basic numerosity perception. These findings reveal the pervasive nature of PAD across cognitive domains and shed light on the mechanisms underlying this disparity. By investigating how the human brain perceives and attends to plants, we take concrete steps toward a better understanding of this still poorly understood phenomenon. Recognizing how deeply PAD is embedded in our cognitive architecture can inform more effective strategies to promote plant awareness and support global plant conservation efforts.

## Author contributions

**Conceptualization:** Silvia Guerra, Luca Battaglini.

**Data curation:** Marco Roccato, Carolina Maria Oletto.

**Formal analysis:** Marco Roccato, Carolina Maria Oletto.

**Funding acquisition:** Silvia Guerra, Luca Battaglini.

**Investigation:** Silvia Guerra, Luca Battaglini.

**Methodology:** Silvia Guerra, Marco Roccato, Luca Battaglini.

**Project administration:** Luca Battaglini.

**Supervision:** Silvia Guerra, Luca Battaglini.

**Visualization:** Silvia Guerra, Marco Roccato, Carolina Maria Oletto, Andrea Ghiani, Marco Bertamini, Luca Battaglini.

**Writing – original draft:** Silvia Guerra, Marco Roccato, Carolina Maria Oletto, Marco Bertamini, Luca Battaglini.

**Writing – review & editing:** Silvia Guerra, Marco Roccato, Carolina Maria Oletto, Andrea Ghiani, Marco Bertamini, Luca Battaglini.

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
