## [Decision Letter · Decision Letter 0]

21 Mar 2026

PONE-D-26-07087At the Roots of Plant Awareness Disparity (PAD): Semantic processing and Numerosity PerceptionPLOS One

Dear Dr. Guerra,

Thank you for submitting your manuscript to PLOS ONE. After careful consideration, we feel that it has merit but does not fully meet PLOS ONE’s publication criteria as it currently stands. Therefore, we invite you to submit a revised version of the manuscript that addresses the points raised during the review process.

We look forward to receiving your revised manuscript.

Kind regards,

Giulia Prete

Academic Editor

PLOS One

Journal Requirements:

“This work was supported by the Department of General Psychology (DPG) BATT_BIRD24_01 - Plant Awareness Project: Numerosity, Training and Brain Stimulation – PRID (Interdisciplinary Department Projects) granted to L.B and S.G.”

5. Please note that funding information should not appear in any section or other areas of your manuscript. We will only publish funding information present in the Funding Statement section of the online submission form. Please remove any funding-related text from the manuscript.

6. Please note that your Data Availability Statement is currently missing the repository name and/or the DOI/accession number of each dataset OR a direct link to access each database. If your manuscript is accepted for publication, you will be asked to provide these details on a very short timeline. We therefore suggest that you provide this information now, though we will not hold up the peer review process if you are unable.

7. We note that Figure 1 in your submission contain copyrighted images. All PLOS content is published under the Creative Commons Attribution License (CC BY 4.0), which means that the manuscript, images, and Supporting Information files will be freely available online, and any third party is permitted to access, download, copy, distribute, and use these materials in any way, even commercially, with proper attribution. For more information, see our copyright guidelines: http://journals.plos.org/plosone/s/licenses-and-copyright.

8. We notice that your supplementary figures are uploaded with the file type 'Figure'. Please amend the file type to 'Supporting Information'. Please ensure that each Supporting Information file has a legend listed in the manuscript after the references list.

Additional Editor Comments:

As you can see, two experts accepted my invitation to evaluate your study: one of the Reviewer is fully convinced and support the publication of the manuscript in its present form. The other Reviewers highlights some issues that in my opinion can be addressed by you. In particular, the Reviewer asked to specify the rationale of the exclusion of values outside the 4-12 range, to better explain overdispersion and distribution of data.

Reviewer's Responses to Questions

**Comments to the Author**

1. Is the manuscript technically sound, and do the data support the conclusions?

Reviewer #1: Yes

Reviewer #2: Yes

2. Has the statistical analysis been performed appropriately and rigorously?

Reviewer #1: Yes

Reviewer #2: No

3. Have the authors made all data underlying the findings in their manuscript fully available?

Reviewer #1: Yes

Reviewer #2: No

4. Is the manuscript presented in an intelligible fashion and written in standard English?

Reviewer #1: Yes

Reviewer #2: Yes

5. Review Comments to the Author

Reviewer #1: Its a simple analysis examining whether plants disparately ignored compared to animals. The results are straight froward and clear and their analyses properly conducted with rigorous statistial analysis. The paper should be accepted as it is

Reviewer #2: The authors Silvia Guerra, Marco Roccato, Carolina Maria Oletto, Andrea Ghiani, Marco Bertamini and Luca Battaglini investigated whether PAD affacts numerosity perception and whether the difference in numerosity perception is caused by the green colour of plants or by semantic recognition. For this, three types of experiments were conducted. The first one testing numerosity perception with coloured items, the second one with black and white items and the third one with items turned by 180° to test for semantic recognition. The study participants had two independent tasks for each experiment, the first was to estimate the number of items, the second was a comparison task. They found that PAD affects numerosity perception of plants based on semantic recognition and not on colour perception.

The study is novel and investigates a phenomenon with important social impact. While the partcipant size, the experimental set-up and the statistical approach seem reasonable, I do have some doubts about the participants involved and the statistics that should be addressed in the paper.

Major concerns/ questions:

- Is there any information on the residence of the participants? It is suggested that people growing up in urban settings are more disconnected to nature and are more likely to have a "nature deficit". So I could imagine that they also differ in PAD or numerosity perception of plants. Did you look into this?

- Statistics: The statistical approach seems appropriate, particularly the use of mixed-models for trial-level data and binominal models for accuracy. What I miss is a reason to exclude values outside the 4-12 range - it seems a bit arbitrary to me. Furthermore, the authors don't report whether model assupmtions were checked. I would like to see that the residuals are approximately normally distributed. For the binominal models, did you check for overdispersion?

- I think the results section would benefit from some more context. It would be nice to start each section with a reminder of the leading hypothesis followed by the results.

- Figure 3: In my view the legend is not comprehensive and should include the statistical method. I would also like to see statistical differences marked with an asteriks in the figure itself.

Minor concerns:

- the resolution of the figures is poor

- "across experiment types" vs"versions" is inconsistent wording.

I would also like to point out that I really enjoyed the discussion and that I valued that interventions were addressed.

6. PLOS authors have the option to publish the peer review history of their article (what does this mean?). If published, this will include your full peer review and any attached files.

Reviewer #1: **Yes:** anthony trewavas

Reviewer #2: No

---

## [Author Response · Author response to Decision Letter 1]

16 Apr 2026

Editor comments:

R1. We thank the Editor for raising this important issue. We have revised the manuscript to ensure full compliance with PLOS ONE style requirements. Please refer to the updated version of the manuscript.

R2. We thank the Editor for this remark. The ORCID ID for the corresponding author has been added and validated in the Editorial Manager system.

R3. We thank the Editor for highlighting this important point. We have revised the Funding Information section to ensure consistency and accuracy of the reported grant details.

“This work was supported by the Department of General Psychology (DPG) BATT_BIRD24_01 - Plant Awareness Project: Numerosity, Training and Brain Stimulation – PRID (Interdisciplinary Department Projects) granted to L.B and S.G.” Please state what role the funders took in the study. If the funders had no role, please state: "The funders had no role in study design, data collection and analysis, decision to publish, or preparation of the manuscript." If this statement is not correct you must amend it as needed. Please include this amended Role of Funder statement in your cover letter; we will change the online submission form on your behalf.

R4. We thank the Editor for highlighting this important issue. Please refer to the revised Financial disclosure: “This work was supported by the Department of General Psychology (DPG) BATT_BIRD24_01 - Plant Awareness Project: Numerosity, Training and Brain Stimulation – PRID (Interdisciplinary Department Projects) granted to L.B. The funders had no role in study design, data collection and analysis, decision to publish, or preparation of the manuscript.”

5. Please note that funding information should not appear in any section or other areas of your manuscript. We will only publish funding information present in the Funding Statement section of the online submission form. Please remove any funding-related text from the manuscript.

R5. Funding information has been removed from the new version of the manuscript and is now reported exclusively in the Funding Statement section, as required. Please refer to the updated version of the manuscript.

6. Please note that your Data Availability Statement is currently missing the repository name and/or the DOI/accession number of each dataset OR a direct link to access each database. If your manuscript is accepted for publication, you will be asked to provide these details on a very short timeline. We therefore suggest that you provide this information now, though we will not hold up the peer review process if you are unable.

R6. We thank the Editor for raising this important issue. The dataset used in the present study is publicly available in the Open Science Framework (OSF) repository: https://osf.io/mzpr5/

The repository contains all the data required to replicate the analyses reported in this manuscript. Specifically, the following datasets were used:

The estimation dataset: https://osf.io/mzpr5/files/8f795

The comparison dataset: https://osf.io/mzpr5/files/u4dxs

7. We note that Figure 1 in your submission contain copyrighted images. All PLOS content is published under the Creative Commons Attribution License (CC BY 4.0), which means that the manuscript, images, and Supporting Information files will be freely available online, and any third party is permitted to access, download, copy, distribute, and use these materials in any way, even commercially, with proper attribution. For more information, see our copyright guidelines: http://journals.plos.org/plosone/s/licenses-and-copyright. We require you to either (1) present written permission from the copyright holder to publish these figures specifically under the CC BY 4.0 license, or (2) remove the figures from your submission: You may seek permission from the original copyright holder of Figure 1 to publish the content specifically under the CC BY 4.0 license. We recommend that you contact the original copyright holder with the Content Permission Form (http://journals.plos.org/plosone/s/file?id=7c09/content-permission-form.pdf) and the following text: “I request permission for the open-access journal PLOS ONE to publish XXX under the Creative Commons Attribution License (CCAL) CC BY 4.0 (http://creativecommons.org/licenses/by/4.0/). Please be aware that this license allows unrestricted use and distribution, even commercially, by third parties. Please reply and provide explicit written permission to publish XXX under a CC BY license and complete the attached form.” Please upload the completed Content Permission Form or other proof of granted permissions as an "Other" file with your submission. In the figure caption of the copyrighted figure, please include the following text: “Reprinted from [ref] under a CC BY license, with permission from [name of publisher], original copyright [original copyright year].” If you are unable to obtain permission from the original copyright holder to publish these figures under the CC BY 4.0 license or if the copyright holder’s requirements are incompatible with the CC BY 4.0 license, please either i) remove the figure or ii) supply a replacement figure that complies with the CC BY 4.0 license. Please check copyright information on all replacement figures and update the figure caption with source information. If applicable, please specify in the figure caption text when a figure is similar but not identical to the original image and is therefore for illustrative purposes only.

R7. We thank the Editor for pointing out this issue and apologize for the oversight regarding the inclusion of copyrighted material in Figure 1. We have removed the original Figure 1 and replaced it with a newly created figure that fully complies with the requirements of the Creative Commons Attribution License (CC BY 4.0). The revised figure does not contain any copyrighted material that would conflict with the journal’s open-access policy. Furthermore, we have carefully revised the figure captions to include appropriate source information and attribution, in accordance with the journal’s guidelines (see lines 148-152, p. 7 and 219-223, p. 10; Manuscript without track changes). We have also reviewed all other figures in the manuscript to confirm their compliance with CC BY 4.0 licensing requirements.

8. We notice that your supplementary figures are uploaded with the file type 'Figure'. Please amend the file type to 'Supporting Information'. Please ensure that each Supporting Information file has a legend listed in the manuscript after the references list.

R8. We thank the Editor for the comment and for carefully reviewing the figures. Now figures are part of the main manuscript, not supplementary materials, and all figure legends are complete and correctly placed in the main text as required.

R9. We thank the Editor for the suggestion. The reviewers did not suggest any specific additional references. However, new references have been added in the Discussion section:

Parsley KM, Daigle BJ, Sabel JL. Initial Development and Validation of the Plant Awareness Disparity Index. Nehm R, editor. CBE—Life Sci Educ. 2022;21: 64. doi:10.1187/cbe.20-12-0275

Pany P, Meier FD, Dünser B, Yanagida T, Kiehn M, Möller A. Measuring students’ plant awareness: A prerequisite for effective botany education. J Biol Educ. 2024;58: 1103–1116. doi:10.1080/00219266.2022.2159491

Marmaroti P, Galanopoulou D. Pupils’ Understanding of Photosynthesis: A questionnaire for the simultaneous assessment of all aspects. Int J Sci Educ. 2006;28: 383–403. doi:10.1080/09500690500277805

Fančovičová J, Prokop P. Development and initial psychometric assessment of the plant attitude questionnaire. J Sci Educ Technol. 2010;19: 415–421. doi: 10.1007/s10956-010-9207-x

R10. We have carefully checked the reference list, and we confirm that it is complete and correct. No retracted articles are cited. Additionally, new references have been added in the Discussion section (see point R9).

Reviewers’ comments:

Reviewer 1.

1. Its a simple analysis examining whether plants disparately ignored compared to animals. The results are straight froward and clear and their analyses properly conducted with rigorous statistial analysis. The paper should be accepted as it is

R1. We thank the Reviewer for the positive evaluation of our manuscript and for recognizing the clarity of the results and the rigor of the statistical analyses. We appreciate the recommendation for publication.

Reviewer 2. The authors Silvia Guerra, Marco Roccato, Carolina Maria Oletto, Andrea Ghiani, Marco Bertamini and Luca Battaglini investigated whether PAD affacts numerosity perception and whether the difference in numerosity perception is caused by the green colour of plants or by semantic recognition. For this, three types of experiments were conducted. The first one testing numerosity perception with coloured items, the second one with black and white items and the third one with items turned by 180° to test for semantic recognition. The study participants had two independent tasks for each experiment, the first was to estimate the number of items, the second was a comparison task. They found that PAD affects numerosity perception of plants based on semantic recognition and not on colour perception. The study is novel and investigates a phenomenon with important social impact. While the partcipant size, the experimental set-up and the statistical approach seem reasonable, I do have some doubts about the participants involved and the statistics that should be addressed in the paper.

1. Is there any information on the residence of the participants? It is suggested that people growing up in urban settings are more disconnected to nature and are more likely to have a "nature deficit". So I could imagine that they also differ in PAD or numerosity perception of plants. Did you look into this?

R1. We thank Reviewer 2 for raising this important issue. We agree that participants’ environmental background, particularly whether they were raised in urban or rural settings, may influence their connection to nature, which in turn could affect both PAD responses and numerosity perception of plants. Unfortunately, we did not collect this information, as it was beyond the scope of the present study. However, we fully agree that including such data would enhance the interpretation of the findings. We have now acknowledged this limitation and added it as a direction for future research in the Discussion section (lines 487-492, p. 22; Manuscript without track changes).

2. Statistics: The statistical approach seems appropriate, particularly the use of mixed-models for trial-level data and binominal models for accuracy. What I miss is a reason to exclude values outside the 4-12 range - it seems a bit arbitrary to me. Furthermore, the authors don't report whether model assumptions were checked. I would like to see that the residuals are approximately normally distributed. For the binominal models, did you check for overdispersion?

R2. We thank the Reviewer for raising these important methodological points, which have allowed us to clarify the rationale underlying the response-range filter, to demonstrate the robustness of our findings across alternative exclusion strategies, and to explicitly report diagnostic checks for both Gaussian and binomial mixed-effects models. We address each point in detail below and in the Supplementary materials which are publicly available in the Open Science Framework (OSF) repository: https://osf.io/mzpr5/

Please note that even if the screen shows ‘0 bytes’, the files are present in the repository. Furthermore, a brief clarification has been added to the main text (see lines 157-158, p. 7 and 269-282, pp. 12-13; Manuscript without track changes).

i) We agree that the original 4-12 response-range filter required clearer justification. Regarding the lower bound, the key distinction is between a response of 0 and responses of 1-3. A response of 0 was treated as qualitatively different, as it most likely indicates that the stimulus was not perceived at all (e.g., due to momentary inattentiveness or occasional display delays in the online experiment). In contrast, responses of 1-3 are not inherently implausible as perceptual estimates. However, they fall within the subitizing range. Regarding the upper bound, responses of 10–12 were retained following visual inspection of the response distribution, in order to preserve a symmetric buffer around the presented numerosity range (7-9). Thus, the 4–12 rule maintains a balanced window around the stimulus range while excluding values that are unlikely to reflect genuine perceptual estimates and are more plausibly due to accidental typographical entries.

Fig. S2. Panel figure showing typed-responses across main range-rule method (please refer to the attach file “Response to Reviewer”)

To directly address the Reviewer’s concern, we conducted a comprehensive sensitivity analysis (see Supplementary Material, OSF repository) by repeating the estimation models under several alternative filtering rules: no exclusion, 4-12 (manuscript rule), 4-19, 1-12, and 1-19. The descriptive results (Table S1), model outputs (Table S2), key contrasts (Table S3), and associated figures (Figures S1–3) show that the fully unfiltered model attenuates several effects, consistent with distortion from clearly implausible responses. In contrast, across all filtered variants, the main effects of category, experiment version, and numerosity remained significant, as did the category × experiment version and experiment version × numerosity interactions. This indicates that the main PAD pattern is not driven by the specific 4-12 rule. More permissive upper bounds (e.g., 4-19) led to some attenuation of higher-order interactions, particularly category × numerosity and the three-way interaction, but the core qualitative conclusions remained unchanged. Additionally, we conducted a supplementary analysis using a participant-specific ±3 SD exclusion rule based on trial-wise estimation error. This analysis (Tables S4-6; Figure S4, Supplementary Material, OSF repository) again preserved the main category pattern and key effects, although some higher-order interact

---

## [Decision Letter · Decision Letter 1]

29 Apr 2026

At the Roots of Plant Awareness Disparity (PAD): Semantic processing and Numerosity Perception

PONE-D-26-07087R1

Dear Dr. Guerra,

We’re pleased to inform you that your manuscript has been judged scientifically suitable for publication and will be formally accepted for publication once it meets all outstanding technical requirements.

Kind regards,

Giulia Prete

Academic Editor

PLOS One

Additional Editor Comments:

As you can see, one of the previous reviewers accepted my invitation to review the new version of your manuscript and recommended its acceptance. The other had recommended acceptance in the first round of review, so he/she was not reinvited. I've also read the new version and am happy to approve its publication in its current form. Congratulations!

Reviewers' comments:

Reviewer's Responses to Questions

**Comments to the Author**

1. If the authors have adequately addressed your comments raised in a previous round of review and you feel that this manuscript is now acceptable for publication, you may indicate that here to bypass the “Comments to the Author” section, enter your conflict of interest statement in the “Confidential to Editor” section, and submit your "Accept" recommendation.

Reviewer #2: All comments have been addressed

2. Is the manuscript technically sound, and do the data support the conclusions?

Reviewer #2: Yes

3. Has the statistical analysis been performed appropriately and rigorously?

Reviewer #2: Yes

4. Have the authors made all data underlying the findings in their manuscript fully available?

Reviewer #2: Yes

5. Is the manuscript presented in an intelligible fashion and written in standard English?

Reviewer #2: Yes

6. Review Comments to the Author

Reviewer #2: I thank the authors for addressing all my concerns. With these changes I think the manuscript is ready for publication.

7. PLOS authors have the option to publish the peer review history of their article (what does this mean?). If published, this will include your full peer review and any attached files.

Reviewer #2: No

---

## [Editor Report · Acceptance letter]

PONE-D-26-07087R1

PLOS One

Dear Dr. Guerra,

I'm pleased to inform you that your manuscript has been deemed suitable for publication in PLOS One. Congratulations! Your manuscript is now being handed over to our production team.

Kind regards,

on behalf of

Dr. Giulia Prete

Academic Editor

PLOS One